# Advances in Intraoperative Flow Cytometry

**DOI:** 10.3390/ijms232113430

**Published:** 2022-11-03

**Authors:** Marcos V. D’Amato Figueiredo, George A. Alexiou, George Vartholomatos, Roberta Rehder

**Affiliations:** 1Department of Neurosurgery, Hospital Estadual Mario Covas, Santo Andre 09190-615, Brazil; 2Department of Neurosurgery, Hospital do Coracao, Sao Paulo 04004-030, Brazil; 3Department of Neurosurgery, School of Medicine, University of Ioannina, 45500 Ioannina, Greece; 4Neurosurgical Institute, University of Ioannina, 45500 Ioannina, Greece; 5Hematology Laboratory, Unit of Molecular Biology and Translational Flow Cytometry, University Hospital of Ioannina, 45500 Ioannina, Greece; 6Department of Neurosurgery, Division of Pediatric Neurosurgery, Hospital Santa Marcelina, Sao Paulo 08270-070, Brazil

**Keywords:** intraoperative, flow cytometry, brain tumors, tumor markers, cell cycle, breast cancer

## Abstract

Flow cytometry is the gold-standard laser-based technique to measure and analyze fluorescence levels of immunostaining and DNA content in individual cells. It provides a valuable tool to assess cells in the G0/G1, S, and G2/M phases, and those with polyploidy, which holds prognostic significance. Frozen section analysis is the standard intraoperative assessment for tumor margin evaluation and tumor resection. Here, we present flow cytometry as a promising technique for intraoperative tumor analysis in different pathologies, including brain tumors, leptomeningeal dissemination, breast cancer, head and neck cancer, pancreatic tumor, and hepatic cancer. Flow cytometry is a valuable tool that can provide substantial information on tumor analysis and, consequently, maximize cancer treatment and expedite patients’ survival.

## 1. Introduction

Flow cytometry (FC) is the gold-standard laser-based technique to measure and quantify cell features. It is an indispensable method in basic, translational and clinical research. The utility of cytometry is based on the use of a flow cytometer and specific reagents, including fluorochromes such as fluorescein, and data analysis software [1]. Such a technique enables one to perform rapid cell analysis with high reproducibility.

Flow cytometry has been widely utilized in the detection and analysis of several diseases, including hematological malignancies, where it may advise diagnosis, guide therapy and be used to follow-up on disease progression [1,2,3]. FC consists of a fast and accurate data collection method from a fluid mixture containing cells and/or their particles and can provide information on cellular physiology or pathological conditions [3]. Samples for flow cytometry can be derived from formalin-fixed cells, frozen samples, fresh tissue, and paraffin-embedded tissues, with the only requirement being that an appropriate pretreatment has been performed so that the cells are suspended in a homogeneous mixture of single cells [2].

Cancer is a leading cause of human mortality. Early cancer detection improves treatment and optimizes patient outcomes and survival. Intraoperative flow cytometry (iFC) has recently emerged as a novel method for cancer diagnosis and treatment, originally being applied in central nervous system tumor surgery and, subsequently, in a number of other malignancies. In the present literature review, we present intraoperative flow cytometry as a novel and adjunct method for standard histopathological and frozen diagnosis to analyze tumor samples and tumor margins. The findings from current reports support the further use of iFC for the accurate characterization of cancer biology and tumor margins.

## 2. Intraoperative Flow Cytometry

The life cycle of a cell, from the beginning of its existence till its division into two daughter cells, is most commonly called the cell cycle. The cell cycle includes a harmonized sequence of events that leads to two cells that share the same amount of DNA. To better understand and study the cell cycle, we divide it into different stages or phases, including the G1, S, G2, and M phases. The G1 earlier phase is the stage where a cell “decides” whether to divide or not. If not, it enters a state of low metabolic consumption, called G0, and if yes, it enters the latter phase of G1, where it prepares to duplicate its DNA [2,4]. The next stage is the S phase, known as synthesis, in which the cell copies the chromosomal DNA. The following stage is known as G2, where it prepares to divide. The next stage is the M stage, also known as mitosis and cytokinesis; in mitosis, the cell separates the two copies of nuclear DNA into two newly formed nuclei, and in cytokinesis, the cytoplasm is also divided to form the two daughter cells. As the M phase is completed and the cell division finishes, a new cell cycle starts over (Figure 1).

Flow cytometry provides a valuable tool to assess the DNA quantity of individual cells, which is proportional for the cells in the G0/G1, S, and G2/M phases, following staining of DNA with a fluorescence dye such as propidium iodide. In addition, following comparison with a normal diploid cell, FC may indicate aneuploidy, which holds diagnostic and prognostic significance.

Alexiou GA et al. described the tumor index (TI) as an important parameter investigated through the DNA histogram that correlates with the proportion of cells in the S and G2/M phases and the degree of tumor malignancy [5,6,7]. The TI is calculated as the cumulative percentage of cells in the S and G2/M phases, corresponding to the proliferating cancer cells. In addition, the quantification of an additional index, the DNA index (DI), may indicate the presence of aneuploidy, providing information on the ploidy of cancer and whether an abnormal number of chromosomes are present in cancer cells. The DI is used to assess the DNA ploidy of a tumor sample by comparing the DNA content of the G0/G1 cell population of presumed tumor cells with that of normal (control) cells. Toward this end, the peak value of the integrated fluorescence of the G0/G1 population of normal cells is considered to be DI = 1.0 (diploid cells). DNA ploidy of tumor cells is expressed as a ratio of the peak value of fluorescence intensity of these cells divided by that of the normal G0/G1 cells. Most frequently, normal lymphocytes/monocytes, including peripheral blood mononuclear cells (PBMCs) from the same patient, can be used as the external standard control of DI = 1.0.

## 3. Brain Tumors

Central nervous system tumors are the leading cause of cancer death among females younger than 19 years old and among males younger than 39 years old [8]. Intracranial gliomas are frequent lesions, of which high-grade gliomas are the most malignant primary brain tumor in adults with a median survival of approximately 15 months. Current treatment includes maximal surgical resection, followed by radiotherapy and concomitant temozolomide chemotherapy [9,10].

Maximal safe resection is the mainstay treatment of low- and high-grade gliomas [9]. A comprehensive understanding of tumor anatomy, as well as the location of the eloquent cortex and subcortical pathways, is required to maximize the extent of resection and preserve the neurological status, which affects overall survival. Current techniques of intraoperative tumor resection include intraoperative magnetic resonance imaging (MRI), neuronavigation systems, frozen section evaluation, and the use of 5-aminolevulinic acid (5-ALA) (Figure 2) [11,12,13].

Given the clinical importance of maximal safe resection, several techniques that can guide tumor resection have been developed over the last years. Among them, intraoperative MRI is very useful to maximize tumor resection; however, it is very expensive and time-consuming [12,13,14]. Frozen section analysis has some limitations, including needing a pathologist with expertise and tissue sample quality issues following preparation. Although there are benefits of using 5-ALA in high-grade glioma resection, it has not been proved to be a good method for the diagnosis of low-grade tumors and metastases [13,14,15].

Recently, desorption electrospray ionization-mass spectrometry imaging (DESI-MS) has emerged as a promising tool for the intraoperative analysis of a tumor margin resection [16,17]. Such a technique may distinguish tumor tissue from adjacent normal tissue based on the detection of different lipid and metabolite profiles.

Another promising tool for tumor diagnosis is intraoperative flow cytometry. In an early experimental report by Mesiwala et al., the feasibility of iFC was supported in the analysis of fresh beef-brain tissue samples. An ultrasound aspiration was utilized for the removal and homogenization of pathological tissue from brain tumors, followed by staining and in situ automatic measurements of DNA content [18].

In a report of iFC in human brain tissue, Alexiou G.A. et al. reported normal brain tissue from epilepsy samples that had a G0/G1-phase fraction of 97.1 ± 0.5%, S-phase fraction of 1.7 ± 0.5%, and G2/M-phase fraction of 1.25 ± 0.5% [5,6]. The higher S- and G2/M-phase fractions correlate with neoplastic lesions. Diploid DNA content and a low proliferative index are observed in benign tumors, including meningiomas, pituitary adenomas, low-grade gliomas, and inflammatory lesions [19,20]. More aggressive lesions often present aneuploid populations and/or a higher proliferation index [19,21,22,23,24,25,26,27].

Different intraoperative flow cytometry protocols have been developed recently for brain tumor analysis. Shioyama et al. described a detailed flow cytometry protocol for a tumor DNA analysis within 10 min based on the malignancy index [28]. The Ioannina protocol, based on rapid cell-cycle analysis, categorizes tumor samples in low- and high-grade lesions, tumor margins, and primary central nervous system lymphoma within 6 min (Figure 3) [5,6].

The utility of iFC has also been proved to go beyond diagnosis. Alexiou et al. described the association between the G0/G1 and S phases and tumor prognosis. Patients with a G0/G1 phase lower than 70% and an S phase greater than 6% presented a worse prognosis. Interestingly, such findings were also observed in meningiomas of grades I and II/III in the G0/G1-phase, S-phase, and G2/M-phase fractions [6,7].

In pediatric brain tumors, including ependymomas, medulloblastomas, atypical teratoid/rhabdoid tumors, astrocytoma, and PNETs, a higher G0/G1-phase fraction and significantly lower G2/M-phase fraction were observed in low-grade lesions [5,6,29,30]. Particularly in ependymomas and medulloblastomas, a significant correlation between the S-phase fraction and Ki-67 index was observed.

Shioyama et al. reported the malignancy index (MI) using flow cytometry in patients with high-grade gliomas surgically treated and followed by radiotherapy and temozolomide [28]. The MI is calculated based on the predominance of a cell type on the histogram and determined by the ratio between the number of cells with greater than normal DNA content and the total cell number. In the presence of DNA aneuploidy, the DNA index was calculated as the ratio of the DNA content in the aneuploid cells and the abnormal number of chromosomes to those in G0/G1 cells. The MI values over 26% are correlated with patient survival and IDH1 mutation status [28].

In intracranial meningiomas, flow cytometry analysis is a promising tool in prognosis and diagnosis. The malignancy of these lesions has been associated with proliferative potential and aneuploidy, as well as with cerebral edema [31].

Since FC analysis can quantify the expression of phenotypic markers, such information can assist in tumor diagnosis. For example, the CD45 marker, a glycoprotein expressed in lymphohematopoietic cells, is diagnostic of primary central nervous system lymphoma (PCNSL). In addition, the expression of an additional marker, CD20, in PCNSL is considered to have a prognostic and therapeutic value, in which treatment with rituximab, a monoclonal antibody against the CD20 antigen, leads to improved survival. Therefore, flow cytometry may have an additional role in the therapeutic strategy in PCNSL patients [29]. Another significant marker expressed in the brain is NCAM, also known as the CD56 marker, with its main isoforms (NCAM120, NCAM-140, and NCAM-180) expressed in several brain tumors, including medulloblastomas, gliomas, and ependymomas. The analysis of CD56 expression can be carried out using flow cytometry. According to recent studies, high-grade lesions present a lower CD56 expression than low-grade tumors [5,30].

## 4. Leptomeningeal Dissemination

Flow cytometry has also proved useful in the analysis of cerebrospinal fluid (CSF) tumor dissemination. Leptomeningeal metastases (LM), known as meningeal carcinomatosis, is a diffuse dissemination of tumor cells into the CSF and leptomeninges (Figure 4). Almost 5–10% of cancer patients with solid tumors will develop LM during their lifetime despite treatment [32].

The most common tumors that develop LM occur in patients with melanoma, breast cancer, small-cell lung carcinoma, and non-small-cell lung carcinoma [33,34,35,36]. Consequently, patients with LM develop multifocal symptoms secondary to the spread of tumor cells in the brain, cranial, and spinal nerves. Prognosis in these cases is dismal [37]. Thus, an early LM diagnosis and treatment initiation may prevent irreversible neurological impairment and symptom control [32].

Current methods for LM screening include Gadolinium-enhanced MRI and CSF cytology. The sensitivity of MRI and specificity is 75% and 20–91%, respectively [38]. In patients with LM, malignant cells are found in the first CSF sample in 50–67% of cases [39]. However, the specificity can be as high as 80% in the second sample of CSF. Flow cytometry methods can detect different fluorescent markers simultaneously. Kerklaan et al. showed a 100% sensitivity and specificity for the EpCAM-based circulating tumor cell (CTC) flow cytometry assay for the diagnosis of LM in patients with epithelial primary tumors. The authors showed the sensitivity of the method to be higher than conventional cytology (61.5%) [40].

Patients with undiagnosed LM for solid tumors may benefit from flow cytometry methods. Such a method may reduce the diagnostic uncertainty of LM and unnecessary lumbar punctures can be prevented. Flow cytometry provides information on diagnosis and prognosis in patients with LM, optimizing treatment initiation and improving patient survival and quality of life [41].

## 5. Breast Cancer

In invasive breast cancer, the prognosis is directly related to patient age, histological grade and subtype, tumor size, lymph node status, and mitotic index [42]. In addition to previous features, the status of hormonal receptors including estrogen and progesterone, c-erbB-2 oncogene expression, expression of Ki-67, and other predictors of disease progression have been included as prognostic factors [42,43,44,45].

In patients with invasive breast cancer, DNA ploidy has been shown to be determined in the early stages of carcinogenesis [46]. Chromosomal instability is directly related to aneuploidy, in which there is an abnormal chromosome number. Aneuploidy is also associated with cancer progression and poor prognosis. Intraoperative flow cytometry analysis has the potential to provide information on DNA ploidy [46,47].

Flow cytometry analysis provides information on nuclear DNA quantification, including on the DNA index, % of S-phase fraction, and ploidy status. DNA ploidy is associated with survival, and presence of aneuploidy is correlated with poor prognosis [46]. In addition, an S-phase fraction (SPF) and mitotic activity have proven to present the same prognosis as the lymph node status in breast cancer [46,48].

Lymph node involvement presents often in patients with a higher incidence of aneuploidy than in patients with no lymph node involvement. Furthermore, it has been observed that there is a correlation between DNA diploidy and node-negative tumors.

Breast cancer tumors presenting with a Ki-67 labeling index of 50% or above are highly proliferative or aneuploid [49]. Additionally, the Ki-67 index is directly related to the proportion of cells in the S phase, which combined with mitotic activity, presented the same prognosis as the lymph node status. In addition, the S phase was directly correlated with tumor recurrence [50].

Intraoperative flow cytometry analysis of the DNA status in breast carcinoma from tumor samples is a powerful tool to inform prognosis, which provides information on the percentage of cells in the S phase, a hallmark of tumor aggressiveness [48,50]. In addition, information about aneuploidy correlates directly with prognosis, as most of the cases have presented aneuploidy and distant metastasis [49]. A recent comparative study of intraoperative flow cytometry with cytology and pathology in patients undergoing lumpectomy evaluated 606 samples of margins and tumors corresponding to 99 patients with invasive ductal carcinoma of no special type and invasive lobular carcinoma. The results showed that intraoperative flow cytometry had a 93.3% sensitivity, 92.4% specificity, and 92.5% accuracy. Cytology had an 82.3% sensitivity, 94.6% specificity, and 94.2% accuracy when pathology was used as the gold standard [47]. Thus, intraoperative flow cytometry can safely predict tumor resection margins, lymph node invasion, and tumor prognosis in a near-real-time fashion.

## 6. Head and Neck Surgery

Squamous cell carcinoma of the head and neck is a very aggressive disease with high mortality and morbidity [51,52]. By the time of diagnosis, approximately 50% of cases have reached the advanced stages of the disease.

Tumor margin resection is a prognostic factor for patients with squamous cell carcinoma, thus improving survival rates and progression-free disease rates [53]. Frozen sections of tumor samples show infiltrative tumors in up to 40% of the patients [51].

Recently, fluorescent tumor tracers have proven effective to visualize tumor margins intraoperatively [54,55]. The addition of γ-glutamyl hydroxymethyl rhodamine green and a clinical fluorescence imaging system showed a specificity over 85% and a sensitivity of 80% on frozen tissue samples [56,57].

Intraoperative flow cytometry analysis is a promising tool in the diagnosis of head and neck tumors [57,58,59,60]. In a prospective study, Vartholomatos et al. correlated diploid/aneuploid lesions and tumor malignancy [57]. According to the authors, all aneuploid lesions analyzed by flow cytometry were neoplastic, in which they presented lower G0/G1-phase fractions and higher S-phase and G2/M-phase fractions than non-neoplastic lesions.

A cut-off value of 88% for the G0/G1 phase had a 97.4% sensitivity and 90% specificity for the diagnosis of neoplastic lesions. For the G2/M-phase fraction, the cut-off value was higher or equal to 5% (80% sensitivity and 86.7% specificity) [57]. The cut-off value of the S-phase fraction for the diagnosis of neoplastic lesions was higher or equal to 6% (97.4% sensitivity and 73.3% specificity). A tumor index (S+G2/M) higher than 10% had a 97.4% sensitivity and 90% specificity for the detection of neoplastic lesions [5,60].

Despite the recent advancements in intraoperative tracers and frozen section analysis, flow cytometry is a novel tool for the diagnosis of squamous cell carcinoma in head and neck regions. Such a technique has shown high sensitivity and specificity in the diagnosis of these intraoperative neoplasms.

## 7. Lung Cancer

Lung cancer has the highest incidence of all tumors, in which non-small-cell lung cancer comprises 80% of new cases of pulmonary tumors [61,62]. Complete tumor resection is considered the mainstay of curative treatments, which requires all the following: free resection margins that are microscopically proven, lobe-specific systematic nodal dissection or systematic nodal dissection, no extracapsular extension of the tumor, and the highest mediastinal node removed must be negative [63].

Approximately two-thirds of the patients with lung cancer with a potentially curative tumor resection will present tumor recurrence and/or disseminated disease [62]. According to Huang et al., lung cancer cells may disseminate immediately after surgical resection [64]. Therefore, the 5-year survival rate has been dismal in the past two decades [65,66].

Novel methods have been developed to identify cancer cells disseminated in the circulation at early stages [65]. Among these methods, flow cytometry, immunocytochemical staining of cytocentrifuge slides, and reverse transcriptase-polymerase chain reaction (RT-PCR) are included [1,3,50].

Flow cytometric analysis provides information on up to four different types of molecules simultaneously on a single-cell basis, thus minimizing the false-positive results [65]. The specificity of flow cytometry for lung cancer in the bloodstream was 97%, which is highly specific for the detection of lung cancer cells in circulation [65]. Thus, such a method may provide essential information during intraoperative lung tumor resection, improving prognosis and overall survival.

## 8. Pancreatic Cancer

Pancreatic cancer is considered one of the most aggressive tumors, presenting high morbimortality rates [67]. Overall survival in pancreatic cancer patients is directly related to tumor margin resection. A surgical procedure is the first-line treatment and a potential cure for early stages of cancer [68,69].

Positive margins and metastasis in pancreatic cancer are associated with reduced overall and recurrence-free survival. Currently, frozen analysis is the gold-standard technique for the detection of positive margins [67].

Intraoperative flow cytometry has shown that it can evaluate the percentage of proliferating cells and is an effective method for detecting cancer cells and the status of tumor margins [70]. Such a technique may represent a novel tool in cancer cell detection and margin status.

## 9. Liver Cancer

The most common primary liver tumor is carcinoma hepatocellular (HCC), the fifth most common cancer, which is also considered one of the most common causes of cancer mortality in men. Complete surgical resection is the mainstay treatment for overall survival in patients with liver cancer or hepatic metastatic disease [71,72].

The status of resection tumor margins is used to assess the prognosis and risk of recurrence. Novel studies using intraoperative flow cytometry have the potential to improve surgical resection and, therefore, overall survival.

Intraoperative flow cytometry tumor analysis is based on the quantification of DNA content, also known as ploidy, and cell-cycle distribution for tumor cell analysis and margin evaluation. Using flow cytometry analysis software, one may determine the G0/G1 geomeans, according to the fluorescence peak [73].

As mentioned before, such techniques may identify areas corresponding to proliferating cells (S phase and G2/M cell-cycle phases) and/or cells with altered DNA content.

In addition, the tumor index reflects the proliferative potential of cell analysis in the S and G2/M phases. A tumor index value over 5% is suggestive of cell proliferation [74].

Markopoulos et al., using intraoperative flow cytometry in patients with hepatic carcinoma margin samples, presented similar results as frozen analysis, demonstrating specificity and sensibility of the flow cytometry method [74]. Such a study showed a modified protocol using a touch imprint on a nylon membrane of the hepatic transaction area to obtain cells for flow cytometric analysis. The novel method is used for cell characterization and margin detection during the excision of primary and metastatic hepatic tumors [74].

Circulating tumor cells from tumor tissues are an independent risk factor associated with the prognosis of solid tumors, including prostate cancer, breast cancer, colon cancer, and hepatocellular carcinoma. These cells are a source of tumor recurrence or metastasis. Therefore, patients presenting elevated CTC counts often have a high recurrence rate, poor outcomes, and decreased overall survival after surgery [75].

The number of CTCs is also associated with the presence of microvascular invasion, particularly in patients diagnosed with HCC. The detection of circulating CTCs may provide a therapeutic window before tumor metastasis. Therefore, the early detection of circulating CTCs may be an additional prognostic indicator to conventional TNM staging and may be a marker of early tumor vascular invasion [73].

Liu et al. have described a strong association between CTC counts and the karyoplasmic ratio, the presence of microvascular invasion, and HCC prognosis using imaging flow cytometry [73]. According to the authors, there is a high sensitivity for the diagnosis of microvascular invasion preoperatively, indicating that the new CTC method is a reliable and sensitive method that uses imaging flow cytometry.

Recent studies of the intraoperative use of flow cytometry contribute to the characterization of tumor margins, the prediction of CTCs, and the potential for complete tumor removal. Intraoperative knowledge of tumor metastasis and complete surgical resection could identify the patients who are at a high risk for local recurrence, assist with follow-up, and help improve overall survival [74].

## 10. Other Cancers

Colorectal cancer is the third most common cancer and represents 10% of all cancer diagnoses. Intraoperative flow cytometry has been utilized for the detection of colorectal cancer cells and in potential resection margin evaluations. In a study that included 106 colorectal cancer patients, samples from malignant and normal colon epithelium were taken and analyzed. The results showed that a threshold of 10.5% for the tumor index (S and G2/M cell-cycle phases) predicted the presence of cancer cells with a 91% accuracy (82.2% sensitivity and 99.9% specificity). For the margin assessment in the subpopulation of rectal cancer patients with or without neoadjuvant therapy, the accuracy of intraoperative flow cytometry was 79% and 88%, respectively [76].

Intraoperative flow cytometry has been also utilized for cancer cell identification during surgery for gynecological malignancies. In a recent study, 21 aneuploid cancers were detected following the DNA index calculation. Tumor samples were characterized by a significantly lower percentage of cells in G0/G1, as well as by an increased tumor index [77].

## 11. Conclusions

Advances in intraoperative molecular analysis using flow cytometry have the potential to improve real-time tumor diagnosis. Such a technique is an attractive adjunct to frozen analysis, especially in time-sensitive scenarios such as the intraoperative assessment of oncological procedures, particularly in brain tumor biopsies and tumor margin resections. Flow cytometry analysis is feasible, cost-effective, and, most importantly, highly sensitive in detecting tumor cells, including CSF dissemination. It is a promising technique that provides substantial information on tumor biology, enabling oncologists and surgeons to maximize cancer treatment and expedite patients´ survival. The data from a plethora of malignancies support the further development of iFC and its utility in the operation theater as a valuable tool in surgical oncology.

## Figures and Tables

**Figure 1 ijms-23-13430-f001:**
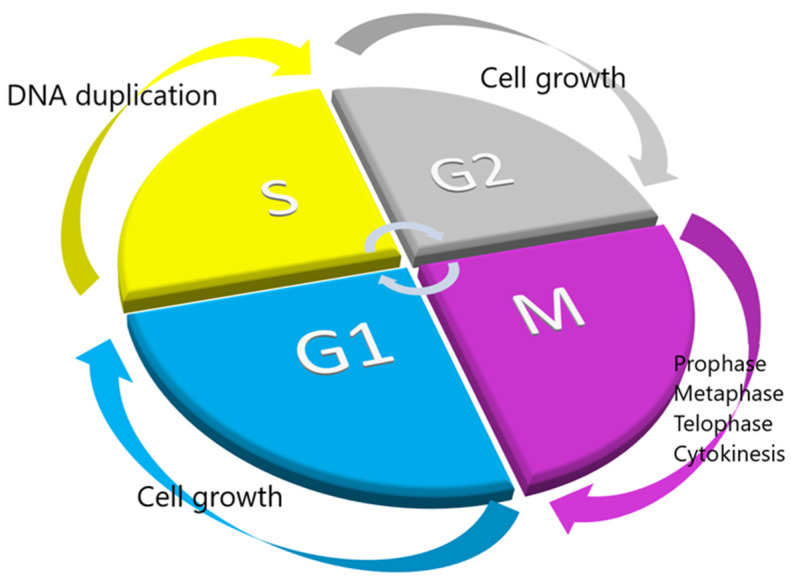
Stages of the cell cycle. The different phases of the cell cycle are presented in a circular fashion to indicate that the end of one cell’s division is the beginning of a new one.

**Figure 2 ijms-23-13430-f002:**
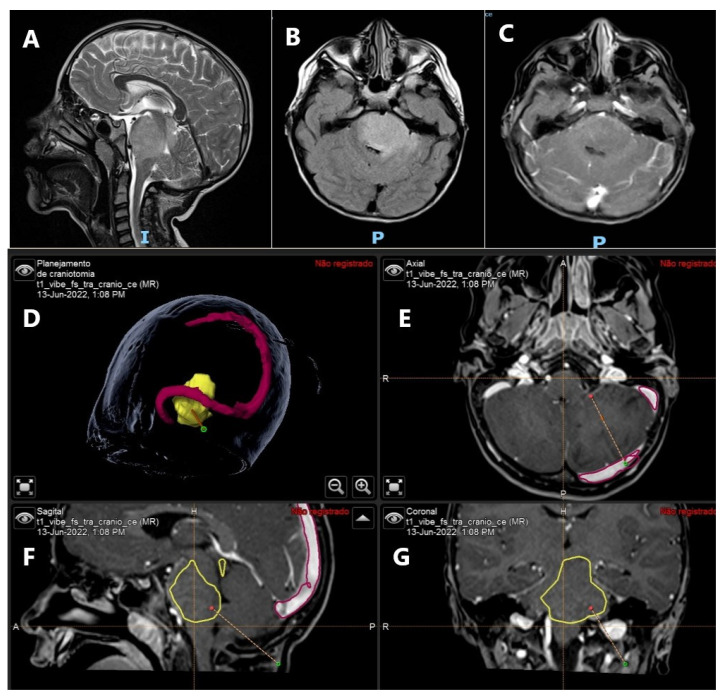
Diffuse intrinsic pontine glioma in a 5-year-old boy. (**A**) Sagittal MRI T2-WI demonstrating diffuse enlargement of the pons. (**B**) Axial T2-Flair image. (**C**) Axial T1-gad showing absence of contrast enhancement. (**D**–**G**) Frameless neuronavigation system planning for tumor biopsy and further molecular study of the tumor. (**D**) Brain lesion is observed in yellow and the transverse and sagittal sinuses in red. (**E**) Axial T1 contrast enhanced. Dotted line showing the planned biopsy trajectory. (**F**) Sagittal T1 contrast enhanced. View of the tumor demarcated in yellow and the tumor biopsy trajectory shown by dotted line. (**G**) Coronal T1 contrast enhanced. View of tumor biopsy trajectory and red dot as the target.

**Figure 3 ijms-23-13430-f003:**
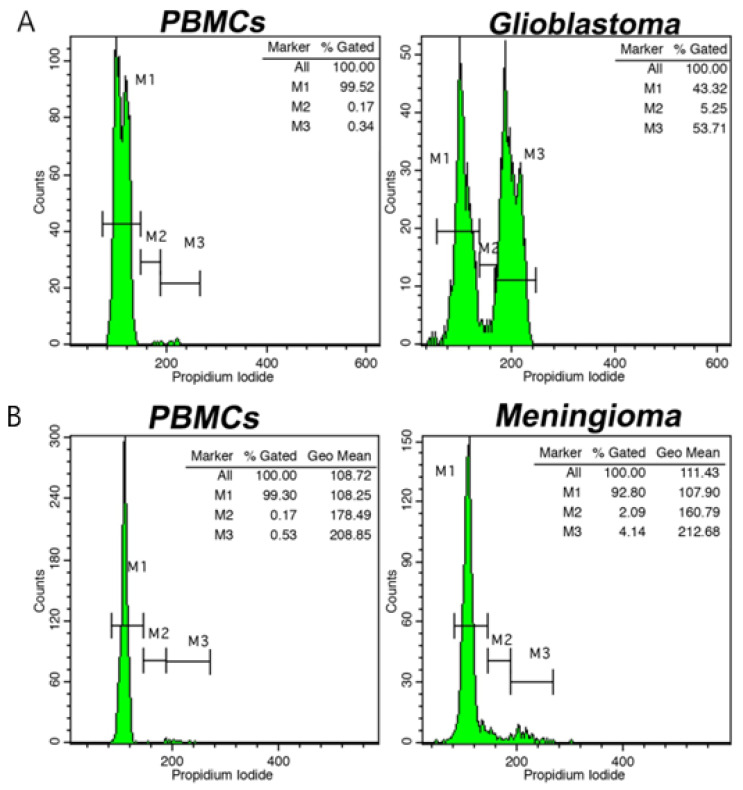
(**A**). Cell-cycle distribution analysis using intraoperative flow cytometry (iFC) in a glioblastoma case (**A**) and in a meningioma case (**B**). Markers M1, M2, and M3 represent G0/G1, S, and G2/M cell-cycle phases, respectively. On the left of each figure, the cell-cycle distribution of peripheral blood mononuclear cells (PBMCs) is presented as control. Tumor index is quantified in each case as the sum of cells from M2 and M3 markers, indicative of proliferating tumor cells. In addition, DNA index (DI) is calculated as the ratio of the geometric mean of cancer cells in M1 (G0/G1 phase) to that of normal cells, in our case, PBMCs. A DI = 1.0 indicates diploid cells; a DI > 1.1, hyperploidy; whereas a DI < 0.9, hypoploidy.

**Figure 4 ijms-23-13430-f004:**
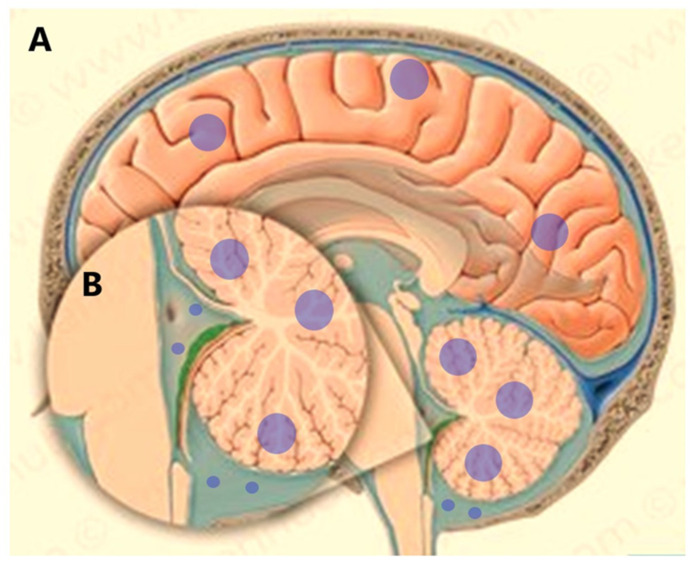
Leptomeningeal dissemination from brain tumors. (**A**). Brain lesions are shown with blue circles. (**B**). Tumor cell dissemination into the CSF space.

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
