# Peer review of "Advances in Intraoperative Flow Cytometry"

_ijms, 2022, doi:10.3390/ijms232113430_

Round 1

Reviewer 1 Report

This review article covers some of the potential uses of flow cytometry as applied to intraoperative analyses of several different cancers.  The manuscript goes through multiple examples of the use of flow cytometry for several different tumor types.  However, the presentation of this by the authors is not a critical analysis, but rather simply a listing of studies that claimed flow cytometry to be useful.  Furthermore, the authors jump back and forth between DNA content analysis and immunostaining without introducing these two distinct methods as such, or describing their limitations.  There also needs to be a fundamental description of how DNA content is determined; e.g., there is no description that a fluorescent DNA stain is used and that fluorescence levels are linearly related to DNA content within individual cells.  Also, there is no good description of what flow cytometry even is to give those who are not familiar with it a clearer picture of how the method works. 

The discussion of Leptomeningeal Dissemination is good and compelling.

The conclusion section is somewhat disappointing in that it does not discuss the barriers to why intraoperative flow cytometry is not used more in surgeries of solid cancers and/or in their treatment.  For example, there would have to be a clinically-certified flow cytometer for use by a hospital for these analyses, and there also would have to be someone trained in its use for this purpose.  Presumably, any procedures to be used during surgery would have to be standardized and approved before coming into common usage.  So, although this reviewer agrees with the authors that flow cytometry of cancer specimens during surgery is promising and would be very useful if standardized methods were developed, there are multiple hurdles to overcome to get to that point.  The authors should discuss this.

In summary, this manuscript correctly makes the point that intraoperative flow cytometry is an underutilized method that would greatly benefit patients.  However, most aspects of the manuscript need to be improved. 

Some specific concerns:

The first sentence in the abstract and Introduction need to be revised.  Arguably, laser scanning confocal microscopy “is the gold standard laser-based technique to analyze and measure cell features.”  It provides details of many more “features” about cells than does flow cytometry, which only measures fluorescence levels in individual cells.  A better sentence would be: “Flow cytometry is the gold standard laser-based technique to measure and analyze fluorescence levels of immunostaining and DNA content in individual cells.” 

Line 29-30: This sentence really does not make much sense.

Line 35-36: This sentence is somewhat misleading because the authors fail to mention that cells need to be single cells in suspension, so that all spatial information in tissues is lost.

Lines 46-52: The description of the cell cycle is oversimplified and somewhat incorrect.

Figure 2: The legend needs more details about panels D-G.

Lines 92-93: Some words must be missing in that sentence. 

Figure 3: The way that the different phases of the cell cycle are identified by the different “M” regions is very crude, subjective, and inaccurate.  There are sophisticated programs, such as ModFit LT, which uses precise algorithms to determine the percentages of cells in each stage of the cell cycle, apoptosis, and cell ploidy.  Such programs should be mentioned in the discussion of the cell cycle, as the analysis in the figure shown is far from state-of-the-art in DNA content analysis. 

Line 146: Analyses of the different m.w. forms of NCAM by flow cytometry would be tricky at best.

Author Response

October 25, 2022

To the Editorial Office, Editor-in-Chief,

Dear Editor,

We were pleased to hear that International Journal of Molecular Sciencesis interested in evaluating a revised version of our manuscript. We are thankful for the very constructive comments that we received on the previous submission. On behalf of the co-authors of the submitted manuscript Ms. No.: ijms-1896201 [ADVANCES IN INTRAOPERATIVE FLOW CYTOMETRY], I have the pleasure of submitting a revised form of this work for consideration.

Based on the reviewer’s comments and suggestions, the manuscript underwent extensive revision and several points in the text have been rewritten, amended, or further analyzed for reasons of precision and clarity. Together with the revised version of the manuscript we are submitting our point-by-point response to the reviewer’s comments.

In the following paragraphs you may find the reviewer’s comments (in blue Courier New font) upon the originally submitted version of the manuscript. You may as well find our response.

Author’s Response to the Review

Reviewer #

1) This review article covers some of the potential uses of flow cytometry as applied to intraoperative analyses of several different cancers.  The manuscript goes through multiple examples of the use of flow cytometry for several different tumor types.  However, the presentation of this by the authors is not a critical analysis, but rather simply a listing of studies that claimed flow cytometry to be useful.  Furthermore, the authors jump back and forth between DNA content analysis and immunostaining without introducing these two distinct methods as such, or describing their limitations.  There also needs to be a fundamental description of how DNA content is determined; e.g., there is no description that a fluorescent DNA stain is used and that fluorescence levels are linearly related to DNA content within individual cells.  Also, there is no good description of what flow cytometry even is to give those who are not familiar with it a clearer picture of how the method works.

The discussion of Leptomeningeal Dissemination is good and compelling.

The conclusion section is somewhat disappointing in that it does not discuss the barriers to why intraoperative flow cytometry is not used more in surgeries of solid cancers and/or in their treatment.  For example, there would have to be a clinically-certified flow cytometer for use by a hospital for these analyses, and there also would have to be someone trained in its use for this purpose.  Presumably, any procedures to be used during surgery would have to be standardized and approved before coming into common usage.  So, although this reviewer agrees with the authors that flow cytometry of cancer specimens during surgery is promising and would be very useful if standardized methods were developed, there are multiple hurdles to overcome to get to that point.  The authors should discuss this.

In summary, this manuscript correctly makes the point that intraoperative flow cytometry is an underutilized method that would greatly benefit patients.  However, most aspects of the manuscript need to be improved.

Some specific concerns:

The first sentence in the abstract and Introduction need to be revised.  Arguably, laser scanning confocal microscopy “is the gold standard laser-based technique to analyze and measure cell features.”  It provides details of many more “features” about cells than does flow cytometry, which only measures fluorescence levels in individual cells.  A better sentence would be: “Flow cytometry is the gold standard laser-based technique to measure and analyze fluorescence levels of immunostaining and DNA content in individual cells.”

ANSWER: We revised as suggested. Abstract:

Flow cytometry is the gold standard laser-based technique to measure and analyze fluorescence levels of immunostaining and DNA content in individual cells.

Line 29-30: This sentence really does not make much sense.

ANSWER: We have revised the sentence.

It is an indispensable method in basic, translational and clinical research. The utility of cytometry is based on the use of a flow cytometer and specific reagents, including fluorochromes such as fluorescein and data analysis software. (1) Such a technique enables one to perform rapid cell analysis with high reproducibility.

Line 35-36: This sentence is somewhat misleading because the authors fail to mention that cells need to be single cells in suspension, so that all spatial information in tissues is lost.

ANSWER: We have revised the sentence.

FC consists of a fast and accurate data collection method from a fluid mixture containing cells and/or their particles and can provide information on cellular physiology or pathological conditions. (3) Samples for flow cytometry can be derived from formalin-fixed cells, frozen samples, fresh tissue, and paraffin-embedded tissues, with the only requirement being that appropriate pretreatment has been performed so that the cells are suspended in a homogeneous mixture single cell.(2).

Lines 46-52: The description of the cell cycle is oversimplified and somewhat incorrect.

ANSWER: We have revised as suggested.

The life cycle of cell, from the beginning of its existence till the division into two daughter cells is most comnly called the cell cycle. The cell cycle includes a harmonized sequence of events that lead into two cells that share the same amount of DNA. To better understand and study the cell cyclem we divide it into different stages or phases, including G1, S, G2, and M phases. The G1 is the stage that in the earlier phase a cell “decides” whether to divide or not. If not it enters a state of low metabolic consumption,called G0, and if yes, it enters the latter phase of G1 where is preparing to duplicate its DNA. (2, 4) The next stage is the S phase, known as synthesis, in which the cell copies the chromosomal DNA. The following stage is known as G2, where the prepares to divide. The next stage is the M stage, also known as mitosis and cytokinesis, where in mitosis the cell separates the two copies of nuclear DNA into two newly formed nuclei and in cytokinesis the cytoplasm is also divided to form the two daughter cells. As the M phase completes and the cell division completes, a new cell cycle starts over. (FIGURE 1)

FIGURE 1. Stages of the cell cycle. The different phases of the cell cycle are presented in a circular fashion to indicate that the end of a cell division is the beginning of a new one

Flow cytometry provides a valuable tool to assess DNA quantity of individual cells, which is proportional of the cells in G0 /G1 phase, S phase and, G2/M, following staining of DNA with a fluorescence dye such as propidium iodide. In addition, following comparison with a normal diploid cell,. FC may indicate aneuploidy which holds diagnostic ans prognostic significance.

Alexiou GA et al. described the Tumor malignancy index (TMI) as an important parameter investigated on the DNA histogram that correlates with the proportion of cells in the S and G2/M phases and the degree of tumor malignancy.(5-7). TI is calculated as the cumulative percentage of cells in S and G2/M phases, corresponding to proliferating cancer cells. In addition, the quantification of an additional index,  DNA index (DI) may indicate the presence of aneuploidy,providing iInformation of the MI includes ploidy of cancer and whether with an abnormal number of chromosomes is present in as well as cancer cells. DI is used to assess DNA ploidy of a tumor sample, by comparing the DNA content of the G0/G1 cells population of presumed tumor cells with that of normal (con-trol) cells. Towards this end, the peak value of the integrated fluorescence of G0/G1 population of normal cells is being considered to be DI = 1.0 (diploid cells). DNA ploidy of tumor cells is expressed as a ratio of the peak value of fluorescence intensity of these cells divided to that of the normal G0/G1 cells. Most frequently normal lympho-cytes/monocytes, including PMBCs from the same patient, can be used as external standard control of DI = 1.0.

Figure 2: The legend needs more details about panels D-G.

ANSWER: We  have revised the figure legend as suggested.

D-G. Frameless neuronavigation system planning for tumor biopsy and further molecular study of the tumor. D. Brain lesion is observed in yellow and the transverse and sagittal sinuses in red.  E. Axial T1 contrast enhanced.  Dotted line showing the planned biopsy trajectory. F. Sagittal T1 contrast enhanced view of the tumor demarcated in yellow and the tumor biopsy trajectory in dotted line.  G.  Coronal T1 contrast enhanced view of tumor biopsy trajectory and red dot as the target.

Lines 92-93: Some words must be missing in that sentence.

ANSWER:  We have revised the sentence as suggested.

Frozen section analysis has some limitations, including the pathologist's expertise and tissue sample quality following preparation. 

Figure 3: The way that the different phases of the cell cycle are identified by the different “M” regions is very crude, subjective, and inaccurate.  There are sophisticated programs, such as ModFit LT, which uses precise algorithms to determine the percentages of cells in each stage of the cell cycle, apoptosis, and cell ploidy.  Such programs should be mentioned in the discussion of the cell cycle, as the analysis in the figure shown is far from state-of-the-art in DNA content analysis.

ANSWER:  We have revised the text and figure as suggested.

FIGURE 3. A. Cell cycle distribution analysis using Intraoperative Flow cytometry (iFC) in a glioblastoma case (A) and in a meningioma case (B). Markers M1, M2, and M3 represent G0/G1, S and G2/M cell cycle phases, respectively. On the left of each figure the cell cycle distribution of peripheral blood mononuclear cells (PBMCs) is presented as control. Tumor index is quantified in each case a the sum of cells in M2 and M3 markers, indicative of proliferating tumor cells. In addition, DNA index (DI) is calculated as the ratio of the geometric mean of cancer cells in M1 (G0/G1 phase) to that of normal cells, in our case PBMCs. A DI = 1.0 indicates diploid cells, a DI > 1.1 polyploidy, while a DI < 0.9 hypoploidy.  

Line 146: Analyses of the different m.w. forms of NCAM by flow cytometry would be tricky at best.

ANSWER:  We have revised as suggested.

Since FC analysis can quantify the expression of phenotypic markers, such information can assist tumor diagnosis. For example, the CD45 marker, a glycoprotein expressed in lymphohematopoietic cells, is diagnostic of primary central nervous system lymphoma (PCNSL). In addition, the expression of an additional marker, CD20 in PCNSL is considered to have a prognostic and therapeutic value, in which treatment with rituximab, a monoclonal antibody against the CD20 antigen, leads to improved survival. Therefore, flow cytometry may have an additional role in the therapeutic strategy in PCNSL patients. (29). Another significant marker expressed in brain, is NCAM, also known as CD56 marker, with its main isoforms (NCAM120, NCAM-140, and NCAM-180) expressed in several brain tumors, including medulloblastomas, gliomas, and ependymomas. The analysis of CD56 expression can be done using flow cytometry. According to recent studies, high-grade lesions present a lower CD56 expression than low-grade tumors. (5, 30)

We would like to thank the reviewers for their helpful decisive comments and suggestions that gave us the opportunity of critically revising our work. We hope this revised version of the manuscript has substantially improved to meet the journal’s standards for publication.

Sincerely,

George Alexiou, MD

Reviewer 2 Report

The article highlights novel clinical trend, describing the specific features and advantages of flow cytometry for intraoperative diagnostics. However  the following improvements are recommended:

- Authors should specify the preferred cell markers suitable for intraoperative flow cytometry with a most significant clinical relevance

- Authors are expected to provide available comparative data regarding the sensitivity and specificity of flow cytometry vs intraoperative histological assessment

- The Conclusion could be extended with an appropriate clinical recommendations. 

Author Response

October 25, 2022

To the Editorial Office, Editor-in-Chief,

Dear Editor,

We were pleased to hear that International Journal of Molecular Sciencesis interested in evaluating a revised version of our manuscript. We are thankful for the very constructive comments that we received on the previous submission. On behalf of the co-authors of the submitted manuscript Ms. No.: ijms-1896201 [ADVANCES IN INTRAOPERATIVE FLOW CYTOMETRY], I have the pleasure of submitting a revised form of this work for consideration.

Based on the reviewer’s comments and suggestions, the manuscript underwent extensive revision and several points in the text have been rewritten, amended, or further analyzed for reasons of precision and clarity. Together with the revised version of the manuscript we are submitting our point-by-point response to the reviewer’s comments.

In the following paragraphs you may find the reviewer’s comments (in blue Courier New font) upon the originally submitted version of the manuscript. You may as well find our response.

Author’s Response to the Review

2) The article highlights novel clinical trend, describing the specific features and advantages of flow cytometry for intraoperative diagnostics. However  the following improvements are recommended:

- Authors should specify the preferred cell markers suitable for intraoperative flow cytometry with a most significant clinical relevance

ANSWER:  In the previous answers of reviewer 1 we have included and discuss several markers apart from cell cycle as suggested.

- Authors are expected to provide available comparative data regarding the sensitivity and specificity of flow cytometry vs intraoperative histological assessment

ANSWER: We have included more data as suggested.

A recent comperative study of intraoperative flow cytometry with cytology and pathology  in patients undergoing lumpectomy evaluated 606 samples of margins and tumors corresponding to 99 patients with invasive ductal carcinoma of no special type and invasive lobular carcinoma. The results showed that intraoperative flow cytometry had 93.3% sensitivity, 92.4% specificity, and 92.5% accuracy. Cytology had 82.3% sensitivity, 94.6% specificity, and 94.2% accuracy when pathology was used as gold standard [47].   Thus, intraoperative flow cytometry can safely predict tumor resection margins, lymph node invasion and tumor prognosis in near real time fashion.

- The Conclusion could be extended with an appropriate clinical recommendations.

ANSWER:  The conclusion section has been revised as suggested.

Advances in intraoperative molecular analysis using flow cytometry have the potential to improve real-time tumor diagnosis.  Such a technique is an attractive adjunct to frozen analysis, especially in time-sensitive scenarios like intraoperative assessment of oncological procedures, particularly in brain tumor biopsy and tumor margin resection.  Flow cytometry analysis is feasible, cost-effective and, most importantly, highly sensitive in detecting tumor cells, including CSF dissemination.  It is a promising technique, which provides substantial information on tumor biology, enabling oncologists and surgeons to maximize cancer treatment and expedite patients´ survival. The data from a plethora of malignancies support the further development of iFC and its utility in the operation theater as a valuable tool in surgical oncology.

We would like to thank the reviewers for their helpful decisive comments and suggestions that gave us the opportunity of critically revising our work. We hope this revised version of the manuscript has substantially improved to meet the journal’s standards for publication.

Sincerely,

George Alexiou, MD

Round 2

Reviewer 1 Report

The manuscript has been significantly improved by the authors' revisions.  However, there are grammatical and spelling errors in some of the revised language.  This needs to be corrected.